# Flow Sensing-Based Congestion Detection for D2D Streaming on a 5G gNB

**DOI:** 10.3390/s22010258

**Published:** 2021-12-30

**Authors:** Chongdeuk Lee

**Affiliations:** Division of Electronic Engineering, Jeonbuk National University, Jeonju-si 54896, Korea; cdlee1008@jbnu.ac.kr

**Keywords:** DUE, 5G wireless mobile, D2D streaming, media flow, channel bandwidth

## Abstract

To provide high-quality streaming services in device-to-device (D2D) communications, performance parameters such as encoding rate, decoding rate, and flow rate should be detected and monitored. The proposed algorithm provides a method to detect time streaming for traffic flows in D2D communications, and a sequence to detect rate imbalance. This paper proposes a new FS-CDA (flow sensing-based congestion detecting algorithm) to prevent high congestion rates and assist an optimized D2D streaming service in 5G-based wireless mobile networks. The proposed algorithm detects and controls flow imbalance for streaming segments during D2D communications, and it includes operations such as transmission rate monitoring, rate adjustment functions, and underflow and overflow sensing for these operations. The paper aims to effectively control traffic flow rates caused by adjacent channel bandwidth, high bit rate error, and heterogeneous radio interference, and to enhance the performance of D2D streaming services by performing such operations. The proposed algorithm for D2D streaming services is measured by deriving the individual weight of certain versions of a streaming flow. Based on the given operations, the simulation results indicated that the proposed algorithm has better performance with respect to average congestion control ratio, PSNR, and average throughput than other methods.

## 1. Introduction

5G technology provides very attractive features along with artificial intelligence, the Internet of Things, and intelligent robot communication in wireless mobile communication services, and this technology both connects and serves various devices, but also through convergence with other devices. It is an important technology that is improving the convenience of life. In 5G mobile communication environments, among the limitations currently experienced by users, the amount of traffic is increasing at a tremendous rate owing to the increase in multimedia and social network services [1,2]. In particular, device-to-device (D2D) streaming services under 5G technologies will become a very important technology as they become generalized, and the service type is expected to converge to streaming-oriented services. 

In 5G-based D2D streaming environments, flow streaming is operated by processing various information requests of D2D user equipment (DUE). In 5G-based wireless mobile networks, DUE and cellular user equipment (CUE) access the wireless network via the base station called the 5G NodeB (gNB), which isconnected by access routers to obtain wireless local area networks (WLANs). For D2D streaming services in this environment, monitoring and detecting large media flow rates are very difficult, and it is an important strategy [3,4].

Such environments cannot cooperatively utilize the resources and channel bandwidth of the gNB. Sensing and managing flow rates at locations close to the mobile clients is an effective strategy for improving the quality of D2D streaming service. In particular, detecting and managing D2D streaming flows at the proxy cache of the gNB is an attractive approach to reducing network traffic congestion and improving the quality of D2D streaming services, owing to the high demand for network bandwidth and continuous playback constraints of D2D streaming applications.

Generally, the upstream traffic from the base station to CUEs is a one-to-many multicast, while the upstream traffic from CUEs to the base station is many-to-one delivery. The forwarding traffic rate from DUE to DUE is one-to-one delivery. Because of the properties of upstream and forwarding traffic, congestion commonly occurs in many-to-one and one-to-one directions. Similarly, the traffic-congestion rates in a gNB can occur in the upstream and D2D direction. The traffic congestion occurring in a gNB is of two types [5,6]. The first type is node-level congestion, which is common in conventional networks. It is caused by buffer flow in the node and can result in packet loss, and increased queuing delay. A packet loss in turn can result in retransmission, subsequently resulting in the consumption of battery power and resource interference. The second type is link-level congestion, which increases packet service time, and decreases both link utilization and throughput; furthermore, it degrades the quality of service (QoS) for DUEs. Both node-level congestion and link-level directly affect D2D streaming QoS.

Therefore, a flow sensing-based congestion detecting scheduler is proposed as an operation to control the flow rates of D2D streaming on a 5G gNB. Existing solutions have entailed algorithms for controlling flow rates such as source rate and peer-to-peer live streaming to solve such problems [7,8]. The sensing-based rate control mechanism under a 5G gNB channel is performed to differentiate flow streaming services despite wireless link errors and resource interference, since the transport protocol must decrease the flow rate only when the traffic in the network experiences rate inconsistency. Algorithms to control the source rate are typically adopted at the DUE layer to optimize the streaming quality, and they are subject to the bandwidth limit provided by the flow control mechanism and QoS requirements for D2D [9,10,11].

Recently, resource allocation and interference mitigation solutions for D2D communications underlaying long term evolution (LTE) cellular networks have primarily been proposed for media streaming in wireless mobile networks [12,13,14]. However, protocols based on resource allocation and interference mitigation have problems such as frequent disconnection and traffic errors, because of their sensitivity to delay and the amount of traffic congestion under 5G applications. In particular, when a D2D streaming service is performed to a gNB channel, the quality of streaming is affected owing to limitations in traffic overflow, limited resources, and bandwidth constraints. In circumstances of limited bandwidth and long delays on a 5G gNB, D2D streaming is affected by radio interference due to traffic congestion and heterogeneous channel bandwidths. The proposed FS-CDA can overcome these constraints and limitations. If a gNB channel does not adequately control flow and traffic congestion caused by D2D streaming, it will degrade the performance of the streaming service, since the packet size of streaming flow is increasing. A control solution for flow rate and sensing-based traffic flow on 5G gNB channels is important for throughput, enhances streaming quality, and supports fairness and reliable responsiveness.

The new FS-CDA for D2D streaming services on 5G gNBs is proposed to enhance the QoS. The proposed mechanism controls congestion by sensing cached traffic flows, as well as the transmission rate, rate adjustment function, underflow, and overflow of these operations. The throughput for a D2D streaming service is improved by considering a rate limitation while supporting fairness and responsiveness of the traffic rate. The proposed mechanism senses and prevents excessive increases in traffic flows in a 5G gNB channel, and the imbalance between rates for DUE and CUE pairs. This operation is crucial for the QoS of D2D streaming. The proposed FS-CDA satisfies the QoS requirement for D2D streaming on 5G gNBs and minimizes rate imbalance owing to the amount of flow congestion and delay in DUE and CUE pairs, and in DUE and DUE pairs. The simulation results show that the proposed mechanism has better performance with respect to packet loss rate, throughput, average response rate, and optimization rate than other methods.

The remainder of this paper is organized as follows. Related work is introduced in Section 2. We introduce the flow sensing-based system model to mitigate congestion and improve the QoS of D2D streaming services in Section 3. The simulation results are evaluated in Section 4, and the concluding remarks are presented in Section 5.

## 2. Related Work

Recently, D2D streaming in the 5G-based wireless mobile networks on 5G gNB has attracted significant interest from both academia and industry as an emerging technology for future global mobile systems [2,12].

Various technologies have been proposed for 5G-based wireless mobile systems, including control of flow and traffic congestion, resource allocation/reuse/sharing, interference-awareness, and power control [15,16,17]. In particular, solutions for flow and traffic congestion contribute significantly to improving quality of experience (QoE) and resource efficiency in the D2D application domain, in addition to fairness and responsiveness of the transport protocol. Typically, the 5G-based D2D streaming suffers from high traffic congestion rates owing to flow error in the buffer cache, imbalances in transmission rate, and restrictions in channel bandwidth and resources. Various solutions have been proposed to solve these limitations [4,16,18].

Xu et al. proposed a solution to mitigate the inference effects between D2D pairs and CUEs to enhance QoS of D2D communications by reusing spectrum resource allocation [19]. This solution can be improved while ensuring quality of experience (QoE) of CUEs by applying possible uplink/downlink resources of the CUEs. To solve the congestion problem caused by interference, this solution should consider a network based on traffic-flow approximation and nonlinear dynamics for both DUEs and CUEs in links.

Lee et al. proposed a solution by applying centralized and distributed algorithms. The centralized algorithm ensures that the CUEs have sufficient probability coverage by preventing interference from D2D pairs [20], and the distributed algorithm ensures that the D2D pairs can maximize the capacity of the spectrum. Although this solution achieves coverage gain, it does not efficiently ensure the QoS for DUEs.

Wang et al. proposed an iterative combinatorial auction, in which the spectrum resource is defined as concepts of resource units for creating bids and the D2D. This solution is an algorithm to improve the sum rate for a D2D underlay network by reducing the interference effect between the D2D pairs and CUEs [21]. Meshgi et al. proposed a resource-allocation algorithm to improve the overall system throughput, while satisfying a particular signal to interference plus noise ratio (SINR) target for both CUEs and D2D groups [22].

Ubaid et al. proposed a multimedia streaming algorithm using D2D in 5G ultra-dense networks [23]. This solution involves scheduling algorithms for streaming the flow content, using D2D communication. In particular, this algorithm ensures the QoS for live video streaming while reducing the flow error for media streaming. However, this solution does not accurately characterize the properties of flow when a high traffic flow rate is transmitted between D2D pairs. To enhance the throughput and fairness, Naderializadeh et al. proposed information theoretic link scheduling [24], and Kim et al. proposed a quantile-based carrier-sense multiple access solution [25]. They focus on optimizing the sum throughput while considering the fairness for DUEs and CUEs.

Meanwhile, to enhance the throughput and fairness, and reduce latency and backhaul link congestion, various caching and equation-based solutions have been proposed [26,27]. The flow traffic in D2D communications is efficient to caching since it requires higher traffic flow rates and provides an imbalanced content reuse property. However, they do not effectively control high flow rates owing to dynamic traffic rates between DUEs and CUEs under a gNB. However, this solution does not accurately characterize the flow when a high traffic flow rate is transmitted between D2D pairs.

However, this solution does not fully encompass the nature of a flow when the traffic flow transmission rate between D2D pairs is high.

To maintain the D2D streaming quality on a 5G gNB, channel properties such as source rate, flow error, and streaming rate should be considered. Nguyen et al. proposed an optimal video streaming solution in a dense 5G network with D2D communications [12]. This solution focused on an optimal rate allocation and description distribution for high-performance video streaming, particularly achieving a high QoE for energy efficiency while restricting co-channel interference over D2D communications in 5G networks. This algorithm optimizes the video segments by characterizing channel state information of the D2D link.

Shanmugam et al. proposed a solution to mitigate the expected downloading time; this solution was an algorithm for video streaming by applying the skewed behavior of users and the cooperation between the base station and small cells [26]. However, this solution does not satisfy the characteristics of wireless channels and encoding techniques of D2D streaming. Huang et al. proposed a solution to solve the problem of balancing the benefit among the cooperators, who seek to join a cooperative video streaming session [27]. This solution avoids unfairness since the cooperators with better downlinks, i.e., higher download rates, expend more cellular traffic and incur higher costs.

To improve the reliability of D2D communications and measure the streaming rate in D2D links, Zhibo et al. proposed a cluster-based interference management algorithm [28,29]. This solution is that a cluster is considered as a frequency reuse for its own D2D pairs. Yaacoub et al. proposed a clustering algorithm for real-time video streaming between DUEs and CUEs on LTE networks [30]. This solution detects the requested videos with consumed energy power and reconstructed distortion. The objective of this is to manage more media flow by taking into account the collaboration between the BS and DUEs. Generally, DUEs and CUEs under an excellent D2D communication environment may enjoy a high-quality media streaming service, while users with poor network service may not. The existing flow control-based solutions does not satisfy QoS for users because the traffic bit rate is unequal to the bandwidth for flow streaming objects.

Recently, various solutions have been proposed to emphasize the importance of accessing multimedia objects with minimal latency, such as text, audio, graphics, images, animation, video, and interactive content [2,12,14]. An efficient method of solving the latency of the D2D streaming in the 5G-based wireless mobile network is to apply D2D communication technology for multimedia delivery.

Therefore, this paper proposes a solution to optimize the conditions of multimedia delivery to satisfy the requirements of D2D streaming in the 5G-based wireless mobile network.

## 3. System Model

In a 5G cellular network, the devices communicate with a base station of 5G core (5G), called the gNB, or a D2D link. Under the 3GPP, the base station of 5G is called the gNB, and the core network is called 5GC [31,32]. In Figure 1, DUE1, DUE2, DUE3, and DUE4 are on the cellular network of gNB1, and DUE5 and DUE6 are on the cellular network of gNB2. In addition, DUE5 and DUE6 communicate through a direct link. Here, the cellular link communicating with the gNB on the uplink system is affected by traffic-flow interference and congestion from D2D users sharing resources, whereas the D2D link is subject to resource interference and congestion from cellular users sharing the same resource block and other D2D users. Figure 1 shows the procedure of traffic flow interference and congestion in the process of sharing resource blocks between the cellular link and the D2D link on gNB1 and gNB2. When DUE3 of the D2D link shares resources with DUE1, it is subject to interference from the cellular link. However, since cellular users DUE1 and DUE4, and D2D users DUE5 and DUE6, are under different gNB channels, they are not affected by interference due to resource sharing.

If they sustain a sufficient distance on gNB1, they are relatively less affected by interference, but if DUE3 and DUE1 do not sustain a sufficient distance, DUE3 is subject to severe traffic flow congestion, owing to resource interference from DUE1.

Therefore, the communication service of the 5G cellular network is optimized by preferentially allocating link resources to each device with high control weight under the gNB.

### 3.1. Detecting Traffic Flow Rates

This section introduces a traffic flow rate detecting function, TFRDF, to detect the congestion due to high traffic flow rates in the DUE buffer cache, and to mitigate the imbalance due to underflow or overflow between DUE and CUE pairs and DUE and DUE pairs.

The TFRDF detects whether the traffic flows are cached with a fixed or variable size in D2D streaming procedure, and whether the burst rate of the traffic flows is long or short. Generally, short traffic flows and burst rates reduce congestion owing to neighboring resource interference, while long traffic flows and bursts do not. The proposed TFRDF detects the flow rates and traffic rates for D2D pairs to reduce congestion due to resource interference, and it has an important function in sensing whether the burst rate of the traffic flows is short or long between DUE and CUE pairs and DUE and DUE pairs.

The congestion flow rate is decided by detecting inconsistencies of link rates between DUE and CUE pairs and DUE and DUE pairs, and it senses the rate imbalance of D2D pairs. Figure 2 shows the TFRDF structure for D2D pairs.

Now, let us calculate the flow rates (*x*) for D2D streaming. Let S(x), R(x) and $DUE_{buffer}(x)$ denote the source rate, caching assign rate, and encoding and decoding flows in the buffer cache between DUEs and DUEs, respectively.

The following notations are also used:TFRDF(x) is the flow rate for D2D streaming.S(x) is the size of the $x^{th}$ flow rate.Burst(x) is the burst rate caching from the buffer cache for $DUE_{buffer}(x)$.B(t) is the burst time received from any DUEs during an arbitrary time stamp *t*.*C*(*x* − *t*) is the occupied capacity of cache.

Let us suppose that the throughput of a D2D streaming increases if no flow error is caused by imbalance. However, in an actual 5G-based wireless mobile channel, the throughput also depends on flow rate, delay, routing failure, and link error in the buffer cache for $DUE_{buffer}(x)$. Therefore, when the flow rates from the buffer cache for $DUE_{buffer}(x)$ are forwarded, *TFRDF*(*x*) is defined as the following.
(1)$$DUE_{encoding}(x)=\{DUE_{encoding}(x-1)+s(x)\}-Burst(x)$$
(2)$$DUE_{decoding}(x)=\{DUE_{decoding}(x-1)+B(t)\}-C(x-t)$$
(3)$$TFRDF(x)=\frac{\sum_{i\in x}\left\{DUE_{decoding}(x-1)-DUE_{encoding}(x)\right\}}{\sum \left\{Burst(x)+B(t)\right\}}\times \mu$$
where *t* is the $DUE_{buffer}(x)$-to-$DUE_{buffer}(x)$ latency during the streaming for a flow rate. *C*(*x* − *t*) is the cache capacity occupied to the buffer cache, and *µ* (0 ≤ *µ* ≤ 1) is the fuzzy value (FV) for a rate adjustment. 

Next, we define the latency after the detection of flow loss. The latency related to the encoding and decoding rates is measured using the transmitted flow rate during an arbitrary time interval. The following equations show the flow delay caused by the encoding rate and decoding rates in the buffer cache.
(4)$$DUE_{buffer}(x+t)=\sum_{k=x+1}^{x=t}\left\{BURST(x)-DUE_{buffer}(x)\}+TFRDF(x\right)$$

If the encoding rate is fixed, the packet loss for flow rates decreases. However, if the encoding rate for flow rates is not fixed or is variable, it suffers from the flow error since the flow rate is mismatched. The flow error owing to overflow or underflow can degrade the QoS for D2D streaming.

The decoding rate is restricted by the buffer cache size of $DUE_{buffer}(x)$, which is frequently restricted in flow rate and size. Therefore, the detection operation to prevent the flow error due to overflow or underflow can be represented by the following Equation (5):(5)$$\sum_{k=x+1}^{x=t}BURST(x)\leq \{E_{r}(k)+D_{r}(k)+TFRDF(x)\}$$
where $E_{r}(k)$ is the encoding rate for an arbitrary flow rate, and $D_{r}(k)$ is the decoding rate.

If Burst(x) in Equation (5) is very large, overflow is expected due to the encoding rate. To prevent this type of flow error, the flow rate between $DUE_{buffer}(x)$ and $DUE_{buffer}(x)$ should be considered.

### 3.2. Detecting Link Flow Errors

If the link flow error in Figure 2 is not controlled effectively, it may cause link congestion owing to the link flow rate between $DUE_{buffer}(x)$ and $DUE_{buffer}(x)$. The link congestion detection operation senses whether the buffer cache for $DUE_{buffer}(x)$ suffers from the link flow error. To effectively detect link flow error, flows for streaming segments are accessed sequentially. After the access, $DUE_{buffer}(x)$ detects the flow rates for $DUE_{buffer}(x)$, …, and $DUE_{buffer}(x+t-1)$ sequentially, and it detects whether the link flow suffers from an error.

#### 3.2.1. Detecting Link Underflow Errors

To detect whether the buffer cache suffers from the link underflow error between $DUE_{buffer}(x)$ and $DUE_{buffer}(x)$, a local cache for $DUE_{buffer}(x)$ detects an encoding rate and a decoding rate. In the local buffer cache, if the decoding rate is less than the encoding rate, the buffer cache suffers from congestion owing to latency and retransmission. Therefore, in order to prevent link problems due to an underflow error between $DUE_{buffer}(x)$ and $DUE_{buffer}(x)$, an FV is assigned in three steps. Step-1 is set to FV ≤ 0.5, and Step-2 involves setting to 0.6 ≤ FV ≤ 0.7. Finally, Step-3 involves setting to FV ≥ 0.8. If the flow source rate between $DUE_{buffer}(x)$ and $DUE_{buffer}(x)$ does not satisfy the link underflow error, S(x) is assigned to the minimum channel bandwidth. If S(x) does not violate the flow source rate, the buffer cache for $DUE_{buffer}(x)$ will not suffer from the congestion due to error in flow source rates.

Therefore, Equation (6) defines $LC_{underflow}(x)$ to prevent the link congestion die to a link underflow error.
(6)$$LC_{underflow}(x)=\sum_{x=0}^{k}\left[\left\{\frac{f(x-t+s(x)}{CBW(x-1)}\right\}+\{E_{r}(k)-Burst(x)\right]$$
where CBW is the channel bandwidth for the flow rate, and $E_{r}(k)$ is the encoding rate for $DUE_{buffer}(x)$.

#### 3.2.2. Detecting Link Overflow Errors

The caching rate is detected to prevent flow error due to mismatched decoding rates. To detect whether $DUE_{buffer}(x)$ suffers from a link overflow error, the flow rate at the decoder buffer is assigned first. The congestion due to link overflow error occurs if the uplinked flow rate is larger than the actual caching rate in the buffer cache. Equation (7) defines $LC_{overflow}(x)$ to prevent the link congestion due to link overflow error.
(7)$$LC_{overflow}(x)=\sum_{x=0}^{k}\left[\left\{\frac{f(x-t+s(x)}{CBW(x-1)}\right\}+\{D_{r}(k)-Burst(x)\right]$$
where $D_{r}(k)$ is the decoding rate for $DUE_{buffer}(x)$.

We can observe that a link overflow error occurs if $E_{r}(k)$ > $D_{r}(k)$, whereas a link underflow error occurs if $E_{r}(k)$ < $D_{r}(k)$. Detecting and controlling link flow errors with source rates is difficult.

### 3.3. D2D Flow Congestion Control

D2D flow congestion control is achieved by sensing the transmission and burst rates between $DUE_{buffer}(x)$ and the requested flow segments. The optimization under the D2D flow operation considers the transmission and burst rates between $DUE_{buffer}(x)$ and the requested flow segments.

#### 3.3.1. D2D Flow Control

D2D flow control which considers the transmission rate in the buffer cache of $DUE_{buffer}(x)$ detects the imbalance between the encoding rate ($E_{r}(k)$ and the decoding rate $D_{r}(k)$). If the operation is violated, the D2D flows suffer from congestion due to transmission rate imbalance. The upper and lower limits of the buffer cache of $DUE_{buffer}(x)$ are set for the optimization of D2D streaming. Equation (8) defines D2D flow control ($DF_{control}(x)$) with consideration of the transmission rate.
(8)$$DF_{control}(x)=Burst(x)-\frac{T-\{(E_{r}(k)-D_{r}(k)\}}{2}\times \mu$$
where *T* is the transmission rate. If the value of *µ* in Equation (8) is small, the variation in the encoding rate becomes large. Hence, the variation in the encoding rate must be considered for D2D flow rate and size.

The encoding rate between D2D pairs for D2D streaming operates a mapping process that converts input data $D_{in}$ to output data $D_{out}$, and is defined as follows:(9)$$D_{out}=f(\mu D_{in}+TV)$$
where TV is a control variable for controlling and managing flow congestion, and f is a non-linear function. The weights of all flows are joined and optimized for a encoding rate. D is input/output data for data transmission between $DUE_{s}$, and the input/output data set includes all stream flows.

During D2D encoding, gNB performs the operation for the congestion and flow error detection, and the operation is defined as follows:(10)$$L(T_{values},{\bar{T}}_{values})=-\log p(T_{values}|{\bar{T}}_{values})$$

Here, the congestion error is defined by a cross entropy loss (CEL) value, and CEL is defined as follows:(11)$$L(T_{values},{\bar{T}}_{values})=-\sum_{k}T_{values}(k)\log({\bar{T}}_{values}(k))+(1-T_{values}(k))\log(1-{\bar{T}}_{values})$$
where $L(T_{values},{\bar{T}}_{values})$ is $P(T_{values}(k))=\frac{1}{{\bar{T}}_{values}(k)}$.

$T_{values}(k)$ represents the target values that affect the DUE in the operation of D2D streaming, and ${\bar{T}}_{values}(k)$ is the target values that do not.

When the encoding rates on the D2D channel effectively are controlled, the optimization of D2D streaming is defined as follows:(12)$$P_{optimization}=\prod_{TV}\arg\min\left[Channel_{T_{values},N,\mu }(L(T_{values},{\bar{T}}_{values}))\right]$$
where $P_{optimization}$ is an optimization function, and N is the additive white Gaussian noise (AWGN) in D2D channel. In this paper, the D2D flow control with the encoding rate is optimized when μ≥0.7.

#### 3.3.2. D2D Flow Control with the Burst Rate

If the transmission rates between the buffer cache of $DUE_{buffer}(x)$ and the requested flow rates are balanced, the D2D streaming due to the transmission rate is optimized. However, the optimization for D2D streaming is not always sustained, since the encoding rate in $DUE_{buffer}(x)$ is variable.

D2D burst rate control: We denote the next flow rate after the streaming of the ${\left(x-1\right)}^{th}$ flow rate as $FR_{next}(x)$. The size of $FR_{next}(x)$ may be fixed or variable. Typically, the rate and size of the flow segment are variable.

Therefore, we consider only variable-size flow segments, which are flows with different burst rates. Here, the encoding rate and bandwidth for a requested rate considers the burst rate. Hence, D2D burst rate control for a requested flow segment ($BUR_{control}(x)$) for $FR_{next}(x)$ in the buffer cache of $DUE_{buffer}(x)$ is defined as follows:(13)$$BUR_{control}(x)\;=\;FR_{next}(x)\times T_{synchro}(x)-\frac{E_{r}(k)}{CBW}\times S(x)$$
where $T_{synchro}(x)$ is the time synchronization for $FR_{next}(x)$ and S(x) is the size of the $i^{th}$ flow segment. The buffer cache capacity for $FR_{next}(x)$ should be considered after the determination of $BUR_{control}(x)$. The buffer cache capacity $BUR_{cache-capacity}(x)$ for $FR_{next}(x)$ is defined as follows:(14)$$BUR_{cache-capacity}(x)=\frac{\left\{E_{r}(k)-D_{r}(k)\right\}}{CBW}\times S(x)\times T_{synchro}(x)+BUR_{control}(x)$$

If $BUR_{control}(x)$ = 0, buffer caching is stopped, resulting in congestion dueto capacity imbalance. Therefore, streaming is performed for the ${\left\lceil \frac{E_{r}}{BW}\right\rceil }^{th}$ flow, with $BUR_{control}\neq 0$ to prevent this type of problem.

Buffer cache control: The flow rates in the buffer cache should be detected continuously to maintain streaming without congestion and latency at $DUE_{buffer}(x)$ to $DUE_{buffer}(x)$.

An imbalance may occur between the rate of a transmitted flow segment and the caching rate. The scenario for $BUR_{control}(x)$ = 0 has already been discussed. Therefore, this scenario should be avoided if possible. Let us consider the following operations:$$T_{synchro}(x)\times T\;>\;S(x)\;\mathrm{and}\;BUR_{control}(x)\;>\;0$$

This is the scenario of increasing bandwidth. It must consider the $T_{synchro}(x)\times T$ operation, since the buffer cache in $DUE_{buffer}(x)$ depends on bandwidth. The encoding rate at the buffer cache of $DUE_{buffer}(x)$ provides better responsiveness. Hence, the encoding rate for $DUE_{buffer}(x)$ is defined as follows:(15)$$BUF_{encoding}(x)=\{S(x)-S(x-1)\}+T_{synchro}(x)\times \mu \times \delta$$
where *δ* (0 < *δ* < 1) is a response coefficient. If $BUF_{encoding}(x)$ = 0, then the detecting operation for the encoding rate is stopped and it is performed from the beginning.

$T_{synchro}(x)\times T$ < S(x) and $BUR_{control}$ < 0

This is the scenario of decreasing bandwidth. Here, the encoding rate ($T_{synchro}(x)\times T$) operation requires the capacity for flow rate, which is operated to *δ*. A larger *δ* provides better responsiveness, and subsequently improving QoS for D2D streaming.

### 3.4. FS-CDA Strategy

We denote the D2D streaming procedure on the cellular link under a 5G gNB as FS-CDA; it schedules the procedure according to the proposed algorithm and guarantees giga-bit based communication streaming for DUEs. In the FS-CDA procedure, DUEs are devices that directly perform D2D communication. In the initial stage of scheduling cellular networking under gNB, CUEs and DUEs sustain weak streaming services with each other, and then gradually sustain a strong service stage. In the D2D streaming procedure, if the FS-CDA scheduler does not know information such as channel status, SINR, and flow error, the link channel of the gNB suffers from traffic congestion; the proposed FS-CDA is a strategy to mitigate such a problem. Therefore, the proposed FS-CDA is an important strategy that improves the QoS of both the cellular and D2D links while overcoming congestion. Pseudo codes arrange the FS-CDA operation and then schedule streaming in the order of the lowest traffic flow error rate and SINR. Thereafter, the scheduler creates the buffer queue $B_{Queue}$ and performs an optimal streaming scheduling operation. At each level, $D2D_{strea\min g}(x)$ for traffic flows, is defined as follows:(16)$$D2D_{strea\min g}(x)=\underset{i,j}{\max}\left\{\frac{1}{1+\left|\mu \times TF_{j}-\mu \times TF_{i}\right|}+TF_{i}^{j}\right\}$$
where $TF_{i}$ is an FV of the $i^{th}$ traffic flow. The max operation is calculated for $i^{th}$ and $j^{th}$ streaming flow in the same pair.

As shown in the Algorithms 1 FS-CDA procedure, DUEs are stably allocated traffic flows to operate an optimal streaming procedure. In this procedure, traffic congestion and delay owing to flow rate errors are mitigated. Therefore, the aim of the proposed FS-CDA is to guarantee streaming QoS by minimizing the latency due to traffic flow error, and procedure codes shows such a scheduling operation. Therefore, FS-CDA effectively operates D2D stream scheduling to ensure a 5G cellular-based streaming service.
**Algorithms 1:** FS-CDAprocedure.Input: Traffic flowsOutput: Traffic flows that satisfy threshold D2D Streaming:BQueue = {TF1,TF2,…,TFn}
//Traffic flows assigned to the buffer queue of DUE∑i=1|BQueue(xi)|≥μTrafficFlowListIndex = 0//Initialization for every traffic flowsif (TrafficFlowListIndex > ∑i=1|BQueue(xi)|≥μ) {D2Dstreaming(x) = maxi,j{11+|μ×TFj−μ×TFi|+TFij}GetTrafficFlowDataSet(TrafficFlowStreamList[FlowStreamListIndex]); //Generation of traffic flow lists satisfying the optimal D2D streaming operation(GetTrafficFlowDataSet(TrafficFlowStreamList[FlowStreamListIndex]), FlowStreamListIndex + 1); //Get the next flow to operate D2D stream scheduling}else if (FlowStreamListIndex < ∑i=1|BQueue(xi)|≥μ) {Threshold = ∑i=1|BQueue(xi)|≥μ//Streaming operation with threshold }else (D2Dstreaming(x)∩∑i=1BCi≠0)//Operation whether continue or stop D2Dstreaming(x){∑i=1|BQueue(xi)|≤0.5//Stop D2Dstreaming(x) operation}

## 4. Simulation Results

In the simulation, the total number of flow objects was set to *N*, and 3 video on demand (VoD) sources were used as standard video clips. The VoD sources had 3000 flow objects extracted from news video and other clips. We created 10 flow blocks from B1 to B10 according to the relationship of detected flow objects. To simplify the simulation, the flow objects were assumed to have a signal-to-noise-ratio and were restricted to a maximum object size of 150 MB.

The other parameters were as follows: the maximum bit stream rate wa 2.55 Mbps; the channel bandwidth was 3.5 GHz in Sub-6; the link bandwidth was 100 MHz in 3.5 GHz, and the mmWave frequency bandwidth was 28 GHz. A cell radius was less than 250 m in our campus area, and the cellular layout was set to 10 cells. The distance from the DUE and DUE, DUE and CUE was randomly distributed from 50 m to 250 m. DUEs and CUEs were assumed to be connected to the access point backbone network, considering the downlink and uplink under gNB. The simulation ran for 580 s with μ≥0.5, 0 < δ < 1, and the time stamp of flow stream $t_{s}$ was [1, 20 s]. The occupied caching rate was less than 90%, and maximum peak signal-to-noise-ratio (PSNR) was less than 35 dB.

For simulation, the PSNR between the original flow object (*i*) and streamed flow object (*j*) was calculated for every combinational of *i* and *j* for i≤j. Considering the flow congestion sensing operation, the distortion reduction $D_{i}$ for *i* can be expressed as follows:(17)$$D_{i}=\{\begin{array}{l} d(i,j)+\sum_{j=2}^{N}d(i,j)\;\;\;\;\;\;(i\;\mathrm{mod}\;N)=1\;\;\;\;\;\\ d(i,j)+\sum_{j=i+1}^{N}d(i,j)-\sum_{j=1}^{N}d(i-1,j),\;otherwise\;\;\;\;\;\;\; \end{array}$$
where d(i,j) is the $D_{i}$ to detect the congestion degree between flow objects *i* and flow *j*, and *N* is the number of total flow objects. Therefore $D_{i}$ can be used to measure the optimal D2D streaming metric and estimate the performance of the client based on successfully received flow objects.

The network model shown in Figure 3 and Figure 4 with one streaming proxy server was used to measure the performance of the proposed algorithm. In the simulation, every flow object from DUEs was uplinked directly to the gNB which checks whether copied objects or flows exist. In the simulation, a generated traffic rate was set to have various sizes from 200 M to 1.5 GB. In the DUEs, the maximum timestamp was set as 20 s.

For the simplicity of simulation, AP1 was assumed to have a DUE client networking model as shown in Figure 4. Thus, a flow object streaming service from SS was transmitted through the path R0-P1-AP1-(DUE1, DUE2)-gNB. However, another path, such as R0-P1-AP2-DUE3-gNB could be used for the flow object streaming service to DUEs. The simulation on the path R0-P1-AP1-(DUE1, DUE2)-gNB indicated a higher packet loss ratio and frequent disconnection, and thus a low throughput if congestion occurs.

We analyzed the results of the proposed algorithm using the simulation parameters shown in Table 1.

We measured the performance while changing the parameters such as the occupied capacity of buffer cache, FV, flow block, number of flow objects, and flow rate. The major metrics used in the evaluation were the average congestion control ratio, PSNR, and average throughput. The metric parameters are the important factors for deciding performance on D2D communications. The proposed algorithm is compared with the other existing schemes: link scheduling algorithm [24], clustering-based algorithm [28], and rate-allocation algorithm [12]. The proposed algorithm was applied with another factor such as FV(μ), cache capacity (CC), and flow object request.

In the first simulation, we analyzed the performance of the average congestion control ratio, PSNR, and average throughput with changing the sizes of flow blocks when the occupied cache capacity CC was 0.5, 0.4, 0.3, and 0.2, respectively, and the response coefficient δ and FV was larger than 0.7, respectively. Figure 5 shows the performance of the average congestion control ratio δ and FV when CC was 0.5, 0.4, 0.3, and 0.2.

Figure 5 shows the simulation result according to the flow block change; when CC was 0.5 the average congestion control ratio had a better performance. However, the results did not improve when CC was lower than 0.5. These results imply that CC and FV affect traffic congestion, and the simulation suggested that they control and manage flow objects effectively.

Figure 6 shows the performance of PSNR when the utilization of CC with δ and FV was 0.9, 0.7, 0.6, and 0.5. A larger CC correlated with a lower flow error ratio and performance improvement, when CC was 0.9. The simulation showed the performance metrics such as D2D streaming quality and flow control given in PSNR. Thus, our proposed algorithm gets better with a higher CC rate among the flow objects.

Figure 7 shows the performance of the average throughput when the utilization of CC was 0.9, 0.7, 0.6, and 0.5, respectively. A larger CC resulted in a higher increase in the throughput, and the efficiency of the excellent performance achieved when the utilization of CC is 0.9.

In the second simulation, we analyzed the performance of the average throughput with flow rate and PSNR with increasing FV to see the effectiveness of sensing control for flow object size and block size. Figure 8 shows the performance of the average throughput when FV was 0.3, 0.5, 0.7, and 0.9. The average throughput showed the best performance when FV was 0.9. This result was because the proposed mechanism applied the TFRDF structure and D2D streaming algorithm based on FV for flow objects.

Figure 9 shows the performance of PSNR with distance when FV was 0.3, 0.5, 0.7, and 0.9. The simulation showed that higher FV effectively controls the flow objects with larger size in cell radius under gNB. As shown in Figure 9, the PSNR results were relatively low because the sizes of all flow objects were set to have a FV. It means that PSNR has little correlation with the distance of the cell radius, and as a result, it has been shown that the quality of D2D streaming is maintained stably.

In the third simulation, we analyzed the performance of the average congestion control ratio, PSNR, and average throughput with increasing FV. Figure 10 shows the simulation result performance with the flow block size ranging from 200 MB to 1.5 GB when FV was 0.1, 0.2, 0.3, 0.4, 0.5, 0.6, 0.7, 0.8, 0.9. As shown in Figure 10, the proposed algorithm exhibited improvement in average performance compared with the link scheduling, clustering, and rate allocation algorithms.

Figure 11 shows the performance of PSNR when FV was 0.1, 0.2, 0.3, 0.4, 0.5, 0.6, 0.7, 0.8, 0.9. The proposed mechanism exhibited an excellent result when FV was 0.9. This meant that a larger FV correlated with a good performance result. This paper has not taken into account the interference constraints to improve the quality of channel bandwidth on gNB. We will consider this parameter in the future.

Figure 12 shows the average throughput performance with flow block sizes ranging from 200 MB to 1.5 GB. We simulated the performance as FV increased and had a better result than others when FV was 0.8 and 0.9.

The proposed algorithm exhibited better results compared with the link scheduling and clustering algorithms, achieving excellent performance. The proposed algorithm is not affected by 5G gNB link constraints, flow characteristics, CC, and other overhead constraints. Hence, the proposed algorithm operates an efficient flow-congestion-control mechanism, and D2D streaming services are maintained in a stable state. As a result, this paper showed that the proposed FS-CDA mechanism has an effect on the performance metrics of the system.

## 5. Conclusions

In D2D streaming environments on 5G gNB, congestion and flow error occur primarily when the network operates large amounts of traffic flows, or the sender for DUE transmits more traffic rates than the receiver for DUE can accept. The congestion and flow delay are responsible for the packet loss and throughput, which degrade the QoS of D2D streaming.

This paper proposes a new FS-CDA mechanism for an optimal D2D streaming service on 5G gNBs to enhance the QoS between $DUE_{buffer}(x)$ and $DUE_{buffer}(x)$. The proposed mechanism is based on the flow trade-off operation between $DUE_{buffer}(x)$ and $DUE_{buffer}(x)$. The mechanism considers the flow rates for encoding and decoding and the traffic-flow-sensing operation for underflow and overflow.

We also considered the flow error and D2D flow control operation due to the transmission rate between $DUE_{buffer}(x)$ and $DUE_{buffer}(x)$, and we maintained the D2D streaming strategy and FS-CDA procedure for operating the optimal D2D streaming operation. A congestion control mapping mechanism and D2D streaming strategy was also proposed to control the traffic flows with reference to FV in the buffer cache. The simulation results indicated that the proposed mechanism outperformed the link scheduling, clustering, and rate allocation algorithms.

## Figures and Tables

**Figure 1 sensors-22-00258-f001:**
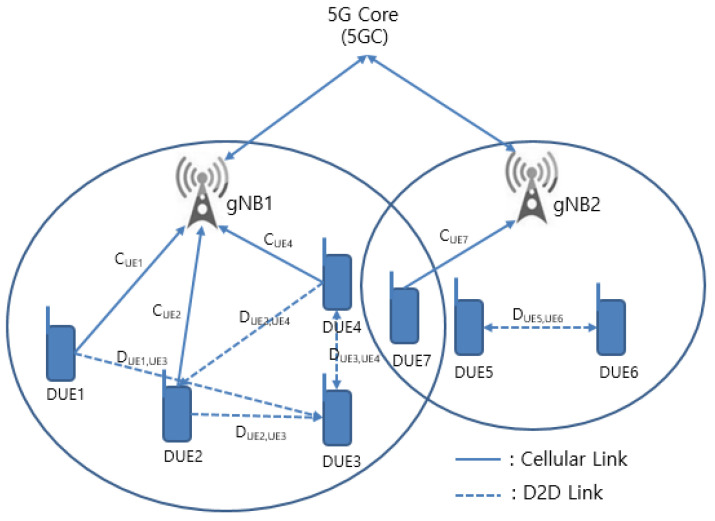
Cellular link and D2D link in 5GC.

**Figure 2 sensors-22-00258-f002:**
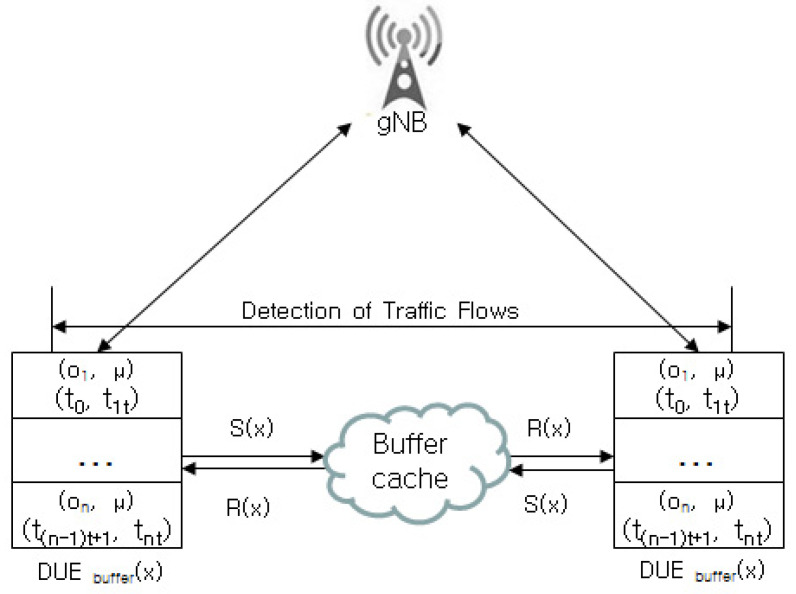
TFRDF structure.

**Figure 3 sensors-22-00258-f003:**
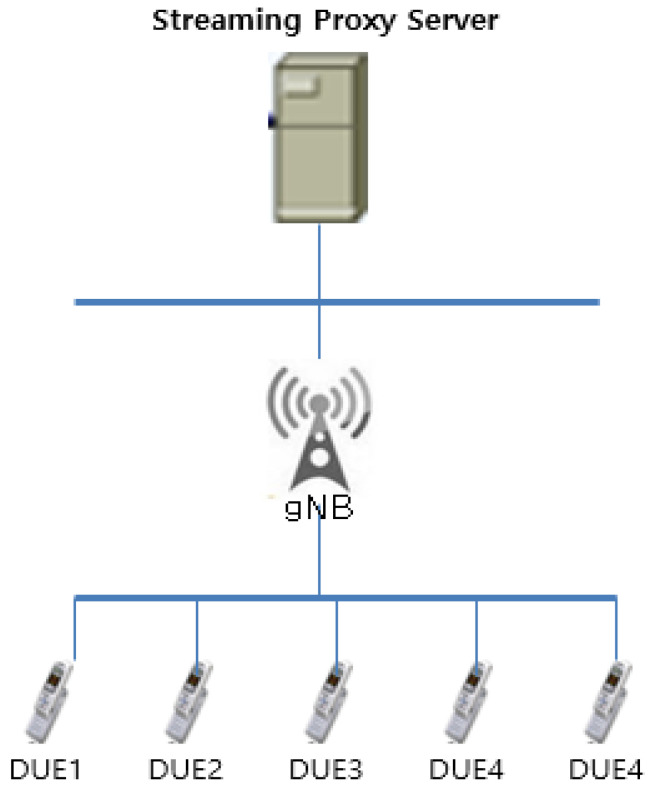
5G-based D2D streaming network model.

**Figure 4 sensors-22-00258-f004:**
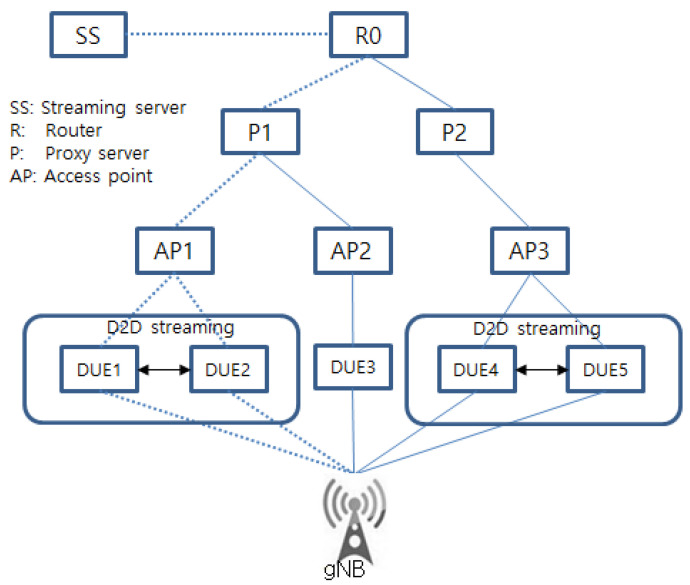
Network model for simulation.

**Figure 5 sensors-22-00258-f005:**
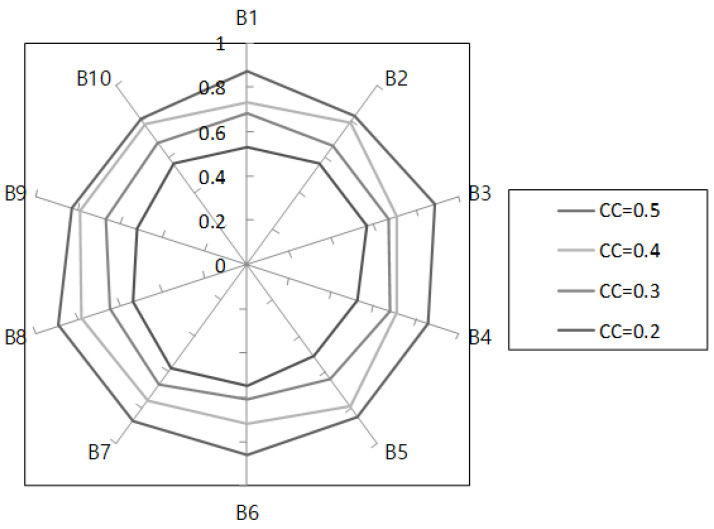
Average congestion control ratio with CC and FV.

**Figure 6 sensors-22-00258-f006:**
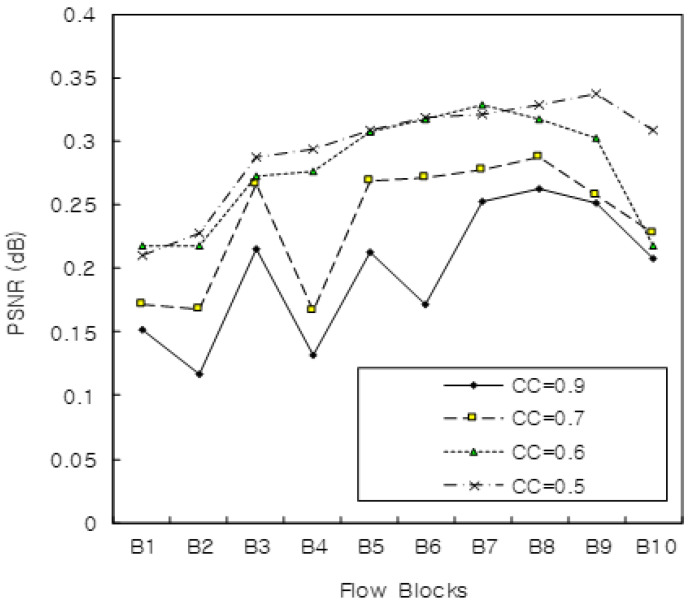
PSNR with flow blocks.

**Figure 7 sensors-22-00258-f007:**
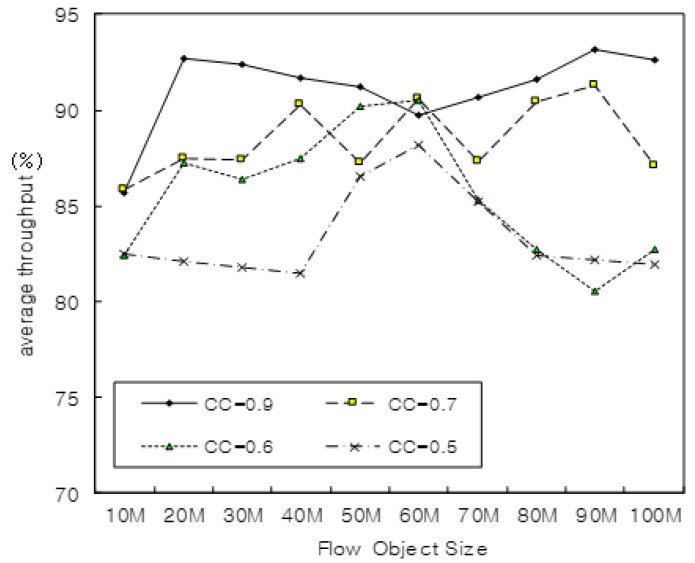
Average throughput with δ and μ.

**Figure 8 sensors-22-00258-f008:**
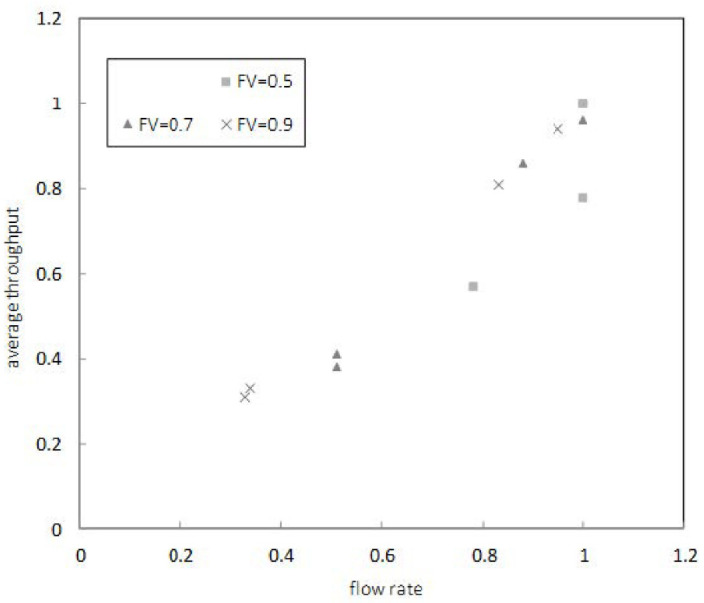
Average throughput with flow rate.

**Figure 9 sensors-22-00258-f009:**
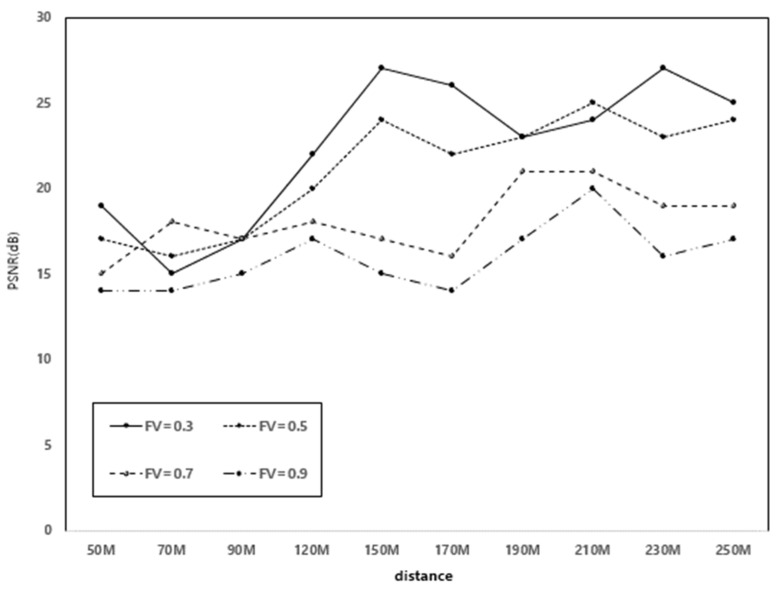
PSNR with distance.

**Figure 10 sensors-22-00258-f010:**
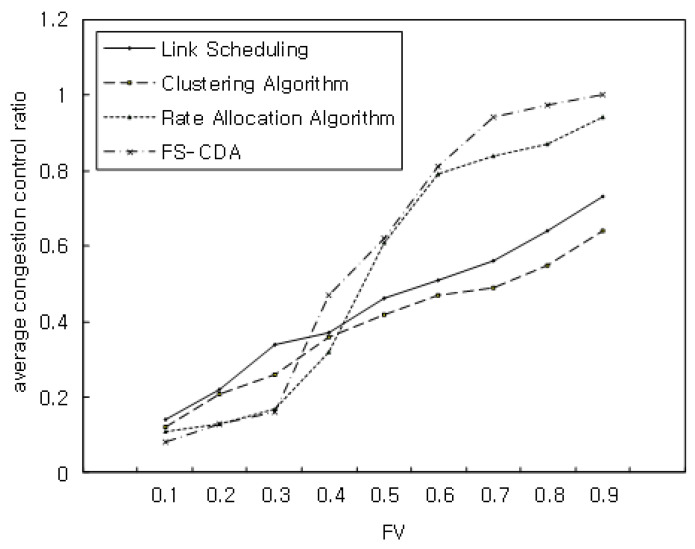
Average congestion control ratio with FV.

**Figure 11 sensors-22-00258-f011:**
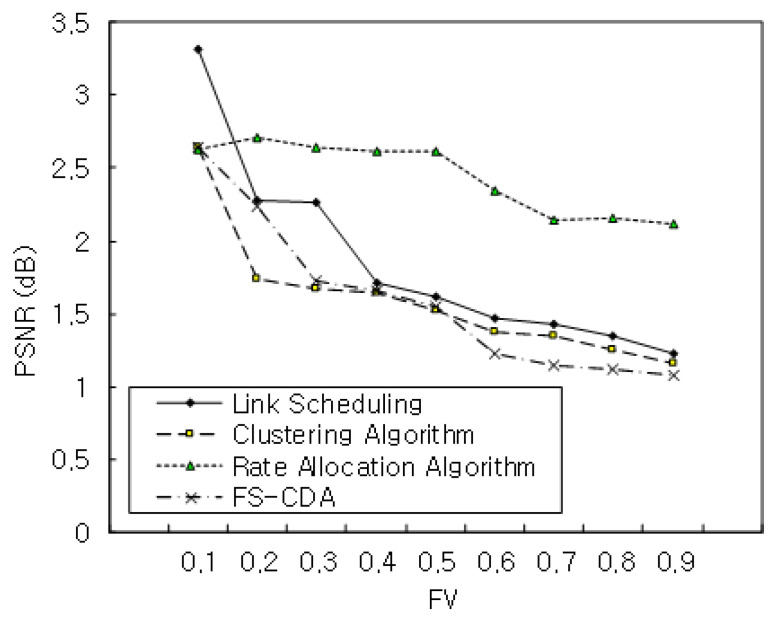
PSNR with FV.

**Figure 12 sensors-22-00258-f012:**
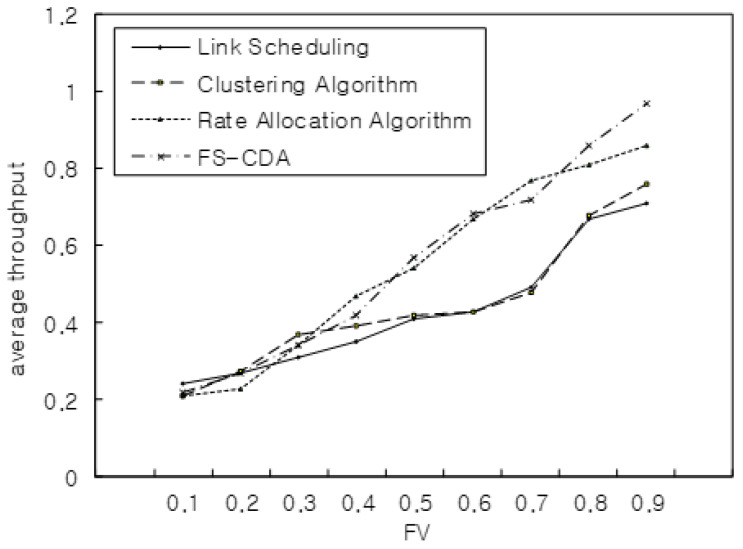
Average throughput with FV.

**Table 1 sensors-22-00258-t001:** Simulation parameters.

Parameters	Value
Total flow objects	3000
The number of flow objects in a block	300–500
Maximum block size	5 Gbytes
Maximum flow object size	250 Mbytes
Total simulation time	580 s
Request time interval	2 s
Time stamp	[1, 20 s]
Buffer cache full	90%
FV	0 ≤ μ < 1
Cell radius	250 m
Link bandwidth	100 Mbps
Noise power	−165 dBm
SINR	10 dB
Average bit stream rate	2.55 Mbps
threshold	μ ≥ 0.5-cut

## Data Availability

Data sharing not applicable.

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
