# Peer review of "Flow Sensing-Based Congestion Detection for D2D Streaming on a 5G gNB"

_sensors, 2021, doi:10.3390/s22010258_

Round 1
Reviewer 1 Report
The paper investigates the Flow Sensing-based Congestion Detection for D2D Streaming. Authors have worked on an area which is recent and interesting. However, there are major revisions needed at this time. My comments are as follows.
- The writeup of the paper needs to be improved considerably. Follow guidelines to the authors as to where the captions of the figures should be placed, how the equations are cited and so on.
- Abstract needs to be shortened. Please follow the instruction to authors of word limit in abstract.
- In line 73, FS-CDMA is mentioned. Please include the reference of it if it is adopted from literature. Also, the acronym is bit strange since in literature CDMA usually refers to code division multiple access. Please clarify.
- In equation (4), the post-script on the summation, i.e., x = t is not clear to me. Please elaborate this in the paper as to what this summation means. Also, the summation variable is not appearing on the LHS after summation in any of the terms.
- In equation (12), the optimization problem is not defined well. I don’t see where is TV in the equation, also N refers to AWGN but I don’t see it in the equation as well. Please follow the standard format of defining an optimization problem.
- Instead of Fig. 3, please include the pseudo-code of the proposed algorithm. At the start of the code, write what are input and what are output of the algorithm.
- All the simulation figures are of very low quality and it is hard to follow the simulation lines and legends. Perhaps better quality figures should be included in the paper.
Author Response
Thank you for your careful reviewing. The authors followed reviewers’ comments and instructions. The response is as follows.
1.The writeup of the paper needs to be improved considerably. Follow guidelines to the authors as to where the captions of the figures should be placed, how the equations are cited and so on.
Response:
The author has poor writeup. Sorry. The author will ask the editor for help to better edit the paper.
- Abstract needs to be shortened. Please follow the instruction to authors of word limit in abstract.
Response:
The author shortened the abstract according to the reviewer's comments and instructions.
Abstract
To provide high-quality streaming service in device-to-device (D2D) communications, performance parameters such as encoding rate, decoding rate, and flow rate should be detected and monitored. The proposed algorithm provides a method to detect time streaming for traffic flows in D2D communications, and a sequence to detect rate imbalance. This paper proposes a new FS-CDMA (flow sensing-based congestion detection monitoring algorithm) to prevent high congestion rates and assist an optimized D2D streaming service in 5G-based wireless mobile networks. The proposed algorithm detects and controls flow imbalance for streaming segments during D2D communications, and it operates operations such as transmission rate monitoring, rate adjustment functions, and underflow and overflow sensing for these operations. The paper aims to effectively control traffic flow rates caused by adjacent channel bandwidth, high bit rate error, and heterogeneous radio interferences, and to enhance the performance of D2D streaming services by performing such operations. The proposed algorithm for D2D streaming services is measured by deriving the individual weight of certain versions of a streaming flow. Based on the given operations, the simulation results indicated that the proposed algorithm has better performance with respect to average congestion control ratio, PSNR, and average throughput than other methods.
- In line 73, FS-CDMA is mentioned. Please include the reference of it if it is adopted from literature. Also, the acronym is bit strange since in literature CDMA usually refers to code division multiple access. Please clarify.
Response:
FS-CDMA is the algorithm proposed by the author. In abstract, the author wrote the full name of FS-CDMA. FS-CDMA is a flow sensing-based congestion detecting monitoring algorithm. Thus, the proposed FS-CDMA is different from CDMA (code division multiplexing access).
- In equation (4), the post-script on the summation, i.e., x = t is not clear to me. Please elaborate this in the paper as to what this summation means. Also, the summation variable is not appearing on the LHS after summation in any of the terms.
Response:
x=t is the transmitted flow rate during an arbitrary time slot. We corrected Equation 4 as follows.
- In equation (12), the optimization problem is not defined well. I don’t see where is TV in the equation, also N refers to AWGN but I don’t see it in the equation as well. Please follow the standard format of defining an optimization problem.
Response:
In Equ.(12), TV is a control variable for controlling and managing flow congestion, and L(Tvalues, overline Tvalues is a flow error detection function. Poptimization(x) was corrected to Poptimization according to the comments.
- Instead of Fig. 3, please include the pseudo-code of the proposed algorithm. At the start of the code, write what are input and what are output of the algorithm.
Response:
FS-CDMA procedure
|
Input: Traffic flows Output: Traffic flows that satisfy threshold D2D Streaming : |
|
= ………-. |
- All the simulation figures are of very low quality and it is hard to follow the simulation lines and legends. Perhaps better quality figures should be included in the paper.
Response:
An author have included simulation Figures with high resolution in paragraph. And We also revised the legend of the Figures.

Reviewer 2 Report
In this paper, the author proposes a method designed to effectively control the traffic flow rate caused by adjacent channel bandwidth, high bit rate errors and heterogeneous radio interference, and perform these operations to improve the performance of D2D streaming media services. In this paper, the proposed algorithm for D2D streaming services is measured by deriving individual weights of certain versions of streaming. Based on the given calculations, the simulation results show that. The algorithm proposed in this paper is better than other methods in terms of average congestion control ratio, PSNR and average throughput. The research results of this paper can provide a basis for comparison, although there will be different applications in special applications, it can be used as a reference direction for high-efficiency design.
In the reviewer’s opinion, in general, the paper is quite interesting. I think the experimental results of this experimental system and the reference values are consistent. The subject of this article contributes to the technological development of industry. Therefore, I think the manuscript can be accepted on "sensors" journal.
Author Response
In this paper, the author proposes a method designed to effectively control the traffic flow rate caused by adjacent channel bandwidth, high bit rate errors and heterogeneous radio interference, and perform these operations to improve the performance of D2D streaming media services. In this paper, the proposed algorithm for D2D streaming services is measured by deriving individual weights of certain versions of streaming. Based on the given calculations, the simulation results show that. The algorithm proposed in this paper is better than other methods in terms of average congestion control ratio, PSNR and average throughput. The research results of this paper can provide a basis for comparison, although there will be different applications in special applications, it can be used as a reference direction for high-efficiency design.
In the reviewer’s opinion, in general, the paper is quite interesting. I think the experimental results of this experimental system and the reference values are consistent. The subject of this article contributes to the technological development of industry. Therefore, I think the manuscript can be accepted on "sensors" journal.
Response:
An author carefully followed the reviewer's comments and instructions.
Thank for your careful reviewing.

Round 2
Reviewer 1 Report
It is suggested to the author to adopt the standards of submitting the review reply. Please resubmit after.
1. Addressing all my previous comments, i.e., if style has been suggested to improve, then improve before resubmission. If captions are asked to be put in format, do so. Avoid using CDMA as an acronym for the proposed scheme and use another acronym if you like, and so on.
2. Highlight all the changes in the paper as well. Even if you put a single dot in the revised version, it should be highlighted so that the reviewer can easily follow the changes made in the paper.
Author Response
We carefully followed reviewer' comments and suggestions.
Thank you.

This manuscript is a resubmission of an earlier submission. The following is a list of the peer review reports and author responses from that submission.
Round 1
Reviewer 1 Report
The Reviewer thinks that this manuscript is not ready for review, as its presentation is very poor. Some comments are shown below.
1. The writing and the use of English grammar need to be improved.
2. Sections 3 and 4 are very difficult to follow.
3. A table listing all symbols is suggested.
4. Did the author show the results of delay?
5. The conclusions section is poorly written.
Reviewer 2 Report
PLease revise the entire article in terms of english langiage, in many points it is not very clear what you mean at a first glance, it has to be understood from the context.
For example:
Page2:
"Due to the properties of upstream and forwarding traffic, the congestion rates commonly occurs in many-to-one and one-to-one direction."
"Pack loss in turn can lead to retransmission and therefore leads to the consumption for battery power and resource interference."
The examplest can continue throughout the paper, it is barely readable. Please find a native speaker and review the entire text.
The mathematical model and the simulation are interesting, there is though a certain level of similarity with an already published work from the same author:
https://www.mdpi.com/1424-8220/21/7/2409
Please underline also the differences with this paper.
Furthermore, it is not clear how you use the adjacent channel bandwidth within the simulations. You are stating "The goal of this paper is to effectively control traffic flow rates caused by adjacent channel bandwidth", I do not find mentioned in the model and in the simulation specifically the use, or influence of the adjacent channel.